# Antiviral Activities of Quercetin and Isoquercitrin Against Human Herpesviruses

**DOI:** 10.3390/molecules25102379

**Published:** 2020-05-20

**Authors:** Chae Hyun Kim, Jung-Eun Kim, Yoon-Jae Song

**Affiliations:** 1Department of Life Science, Gachon University, Seongnam-Si, Gyeonggi-Do 13120, Korea; cogus9504@naver.com (C.H.K.); jeunkim17@gmail.com (J.-E.K.); 2Program in Developmental and Stem Cell Biology, The Hospital for Sick Children, Toronto, ON M5G 0A4, Canada

**Keywords:** varicella-zoster virus, human cytomegalovirus, antiviral, quercetin, isoquercitrin

## Abstract

We previously reported that the ethyl acetate (EtOAc) fraction of a 70% ethanol extract of *Elaeocarpus sylvestris* (ESE) inhibits varicella-zoster virus (VZV) and human cytomegalovirus (HCMV) replication in vitro. PGG (1,2,3,4,6-penta-O-galloyl-ß-D-glucose) is a major chemical constituent of the EtOAc fraction of ESE that inhibits VZV but not HCMV replication. In this study, we comprehensively screened the chemical compounds identified in the EtOAc fraction of ESE for potential antiviral properties. Among the examined compounds, quercetin and isoquercitrin displayed potent antiviral activities against both VZV and HCMV with no significant cytotoxic effects. Both compounds strongly suppressed the expression of VZV and HCMV immediate–early (IE) genes. Our collective results indicated that, in addition to PGG, quercetin and isoquercitrin are bioactive compounds in the EtOAc fraction of ESE that effectively inhibit human herpesvirus replication.

## 1. Introduction

Herpesviruses cause various diseases in humans and animals [1]. The virus consists of a large DNA genome within the icosahedral capsid surrounded by tegument proteins and a lipid bilayer composed of several viral and glycoproteins [2]. Human herpesviruses are divided into three subfamilies (alpha, beta, and gamma) on the basis of their biological and molecular properties [3].

The human herpesvirus undergoes two life-cycle phases, lytic replication and latency [1]. During lytic replication, the virus executes the lytic-gene cascade involving the coordinated expression of lytic immediate–early (IE), early (E), and late (L) genes. Immediate–early genes encode transactivators of viral and cellular genes to create an optimal cellular state for viral DNA production. Early genes encode proteins that facilitate viral DNA replication, and late genes encode viral structural proteins.

Previously, we reported that the ethyl acetate (EtOAc) fraction of a 70% ethanol extract of *Elaeocarpus sylvestris* (ESE) inhibits varicella-zoster-virus (VZV) and human-cytomegalovirus (HCMV) replication in vitro [4,5]. The EtOAc fraction of ESE contains several chemicals, such as luteolin-7-rutinoside, isoquercitrin, quercetin-3-O-arabinoside, luteolin-4-O-glucoside, quercetin, galloyl-D-glucose, digalloyl glucose, gallic acid, digallic acid, trigalloyl glucose, tetragalloyl glucose, ellagic acid and 1,2,3,4,6-penta-O-galloyl-ß-D-glucose (PGG). From these, PGG is a major constituent [6] and was recently characterized as a potent inhibitor of VZV [5].

VZV, a member of the alpha herpesvirus family, is transmitted through aerosols or direct contact with the virus in lesions and infects the respiratory mucosal epithelium [7]. Primary infection of VZV causes chickenpox (varicella) in young children and establishes latent infection in dorsal root ganglia. Reactivation of VZV from latency can cause shingles (herpes zoster) [7]. In cell culture, VZV is highly cell-associated and spreads via cell-to-cell contact [8]. HCMV, a member of the beta herpesvirus family, is transmitted via physical contact, breastfeeding, blood transfusion, or organ transplantation [9]. Similar to VZV, HCMV establishes a lifelong latent infection with periodic reactivation after primary infection [10]. Primary infection and reactivation of HCMV is usually asymptomatic in healthy individuals [9]. However, HCMV can be fatal in immunosuppressed or immunocompromised individuals such as organ-transplant recipients or AIDS patients [11]. In addition, HCMV infection during pregnancy has been associated with infant morbidity, childhood hearing loss, and other neurodevelopmental defects [12,13].

Commercially available antivirals to treat VZV and HCMV infections include acyclovir (ACV) and ganciclovir (GCV), respectively [14]. Both are nucleoside guanosine analogs that are activated by viral thymidine kinases to form nucleoside triphosphate, and they interfere with viral DNA polymerase activity [15]. Although ACV and GCV effectively inhibit herpesvirus replication, side effects and toxicity are major concerns [16,17]. Moreover, the emergence of viral strains resistant to ACV and GCV poses a significant public-health challenge [18], highlighting the urgent need to develop alternative antiviral therapies against herpesviruses. Since PGG exerts antiviral effects against VZV but not HCMV, we focused on the antiviral effects of chemical constituents of the EtOAc fraction of ESE against both viruses in this study with a view to identifying additional bioactive compounds.

## 2. Results

### 2.1. Antiviral Activities of Chemical Compounds Identified in EtOAc Fraction of ESE against VZV and HCMV

Ten out of thirteen compounds identified in the EtOAc fraction of ESE that were commercially available were screened for potential anti-VZV and anti-HCMV activity (Figure 1) [5]. Among the examined compounds, quercetin and isoquercitrin (quercetin 3-*O*-β-D-glucoside) significantly inhibited VZV and HCMV replication (Figure 1). Compared to VZV-infected human-foreskin-fibroblast (HFF) cells treated with DMSO, the relative amount of viral DNA was reduced to 19.7% and 34.1% in VZV-infected HFF cells treated with quercetin and isoquercitrin, respectively (Figure 1A). Additionally, quercetin and isoquercitrin treatment led to a reduction in the relative abundance of HCMV DNA to 32.2% and 20.6%, respectively (Figure 1B). Ellagic acid (EA) exhibited antiviral activities against both VZV and HCMV, but exerted a significant cytotoxic effect on HFF cells (Figure 1 and data not shown). As reported previously, 1,2,3,4,6-penta-O-galloyl-ß-D-glucose induced a significant decrease in VZV replication (Figure 1A) [5], but had no antiviral activity against HCMV (Figure 1B). On the basis of these results, we further investigated the effects of quercetin and isoquercitrin on VZV and HCMV replication (Figure 2).

### 2.2. Antiviral Activities of Quercetin and Isoquercitrin Against VZV and HCMV

To determine the antiviral activities of quercetin and isoquercitrin, a plaque-reduction assay was performed. The average 50% inhibitory concentrations (IC_50_) of ACV for VZV and GCV for HCMV were 3 and 0.89 µg/mL, respectively [21,22]. Quercetin exhibited potent antiviral activities against both VZV and HCMV, with estimated IC_50_ values of 3.835 ± 0.56 and 5.931 ± 1.195 µg/mL, respectively (Table 1). Isoquercitrin exhibited significant antiviral activity against HCMV, with an IC_50_ value of 1.852 ± 1.115 µg/mL, but was less effective than quercetin against VZV (IC_50_ of 14.4 ± 2.77 µg/mL) (Table 1).

To examine whether antiviral activities of quercetin and isoquercitrin were due to cytotoxic effects against host cells, HFF cells were treated with the compounds at concentrations of 0, 10, 20, 50, and 100 µg/mL, and cell viability was determined by measuring cellular ATP levels at 24, 48, and 72 h after treatment (Figure 3). Consistent with previous reports, cellular ATP levels in control HFF cells (0 µg/mL) increased during the incubation period (24 to 72 h), possibly due to proliferation (Figure 3) [2]. Compared to the 0 h time point, cellular ATP levels were not reduced in HFF cells treated with 10 and 20 µg/mL quercetin (Figure 3A). At 72 h after quercetin treatment (20 µg/mL), proliferation was reduced to 63.3% (Figure 3A). Isoquercitrin treatment (10 and 20 µg/mL) had no effects on cellular ATP levels or proliferation (Figure 3B). However, at a treatment concentration of 50 µg/mL, isoquercitrin reduced cellular ATP levels and exhibited antiproliferative activity (Figure 3B). At 72 h after treatment, the IC_50_ values of quercetin and isoquercitrin for cell proliferation were determined as 38.6 ± 5.31 and 73.83 ± 21.98 µg/mL, respectively. Since the IC_50_ values for antiviral activity of quercetin and isoquercitrin did not correspond to those for antiproliferative activity, we concluded that quercetin and isoquercitrin interfere with VZV and HCMV replication without exerting significant cytotoxicity.

### 2.3. Quercetin and Isoquercitrin Inhibits VZV and HCMV Lytic-Gene Expression

To further determine the effects of quercetin and isoquercitrin on lytic-gene expression, VZV- and HCMV-infected HFF cells were treated with quercetin and isoquercitrin, and IE, E, and L transcript levels were determined using qRT–PCR (Figure 4 and Figure 5). The levels of VZV *ORF62* (IE), *ORF28* (E), and *gB* (L) transcripts [23] were significantly reduced in the HFF cell group treated with quercetin and isoquercitrin (Figure 4). As reported previously, PGG suppressed VZV lytic-gene transcript levels (Figure 4) [5]. Quercetin and isoquercitrin treatment additionally reduced the transcript levels of HCMV *UL122* (IE), *UL44* (E), and *UL83* (L) [24] (Figure 5). Interestingly, PGG had a less significant effect on HCMV lytic-gene expression than that of quercetin and isoquercitrin (Figure 5).

To further ascertain the effects of quercetin and isoquercitrin, protein levels of VZV IE62 (encoded by *ORF62*) and HCMV IE2 (encoded by *UL122*), which are essential for viral lytic-gene expression, were determined via Western blot analysis [23,24] (Figure 6). Consistent with qRT–PCR data, the protein levels of both VZV IE62 and HCMV IE2 were significantly reduced in the presence of quercetin and isoquercitrin (Figure 6, compare Lanes 4 and 5 with Lane 2). As expected, PGG suppressed the expression of the VZV IE62 but not the HCMV IE2 protein, consistent with its antiviral activity specifically against VZV and not HCMV (Figure 6, compare Lanes 6 and 2). Our results clearly indicated that quercetin and isoquercitrin significantly interfere with VZV and HCMV lytic-gene expression.

## 3. Discussion

Potent antiviral activities of the EtOAc fraction of ESE against human herpesviruses, including VZV and HCMV, were recently documented [4,5]. To date, 13 chemical components of this fraction were identified, among which PGG is a major constituent [6]. PGG inhibits VZV replication, but exerts no significant effects on HCMV replication [5], suggesting that other chemicals in the EtOAc fraction contribute to HCMV inhibition. Another possibility is that chemical components other than PGG also exhibit anti-VZV activity. Further analysis of the EtOAc fraction of ESE in the current study led to the identification of quercetin and isoquercitrin as potential antiviral (anti-VZV and anti-HCMV) compounds (Figure 1).

Quercetin and isoquercitrin are flavonoids that exhibit various biological activities [25,26]. Quercetin is the most abundant dietary flavonoid, with reported antioxidant, anti-inflammatory, antihypertensive, antimicrobial, antiatherogenic, and anticancer effects [25]. The antiviral activity of quercetin against HCMV was also demonstrated [27]. Isoquercitrin, also known as quercetin 3-*O*-β-D-glucoside, is a glucoside derivative of quercetin with enhanced bioavailability [28] that exerts chemoprotective effects against oxidative stress, cancer, cardiovascular diseases, diabetes, and allergic reactions [26].

The antiviral activities of quercetin and isoquercitrin against herpes simplex virus type 1 (HSV-1) and 2 (HSV-2) were extensively reported [29,30,31,32,33,34]. Quercetin and isoquercitrin suppress NF-κB activation in HSV-1-infected cells [31,33]. Quercetin was also reported to reduce HSV-1-induced interferon (IFN) regulatory factor 3 (IRF3) activation [33] or to inhibit HSV-1 entry into host cells [31]. By performing pre- and post-treatment studies, quercetin was reported to inhibit HSV-1 infection in the early stages of the viral life cycle (from 0 to 2 h postinfection) [30].

In agreement with findings in a previous study [30], quercetin and isoquercitrin treatment at 24 and 48 h after infection exhibited no significant antiviral effects against VZV and HCMV (data not shown). These results suggest that quercetin and isoquercitrin interfere with the early steps of VZV and HCMV infection. Since VZV is highly cell-associated and spreads via cell-to-cell contact in cell culture [8], quercetin and isoquercitrin are not likely to interfere with VZV entry into host cells.

During the lytic replication of herpesviruses, the expression of IE genes is essential for the sequential cascade of lytic-gene expression and viral DNA replication [3]. Our data suggest that quercetin and isoquercitrin inhibit viral lytic-gene expression and replication through the downregulation of IE genes of VZV (ORF 62) and HCMV (IE2) (Figure 6). The compounds may either directly inhibit the functions of cellular-transcription factors or indirectly interfere with signaling pathway(s) to activate transcription factor(s) that regulate major IE (MIE) enhancer/promoter (MIEP) activation.

Both VZV and HCMV activate the c-jun N-terminal kinase (JNK) pathway. Activator protein 1 (AP1), a downstream-transcription factor of the JNK pathway, is critical for the activation of both VZV and HCMV MIEP [35,36]. One possibility is that quercetin and isoquercitrin inhibit MIEP activation by interfering with the JNK pathway. However, quercetin was reported to inhibit JNK activation, while isoquercitrin exerts the opposite effect [37,38]. In addition, PGG interferes with VZV-induced JNK activation and VZV IE62 expression while having no effect on HCMV IE2 expression. Therefore, mechanisms other than JNK inhibition may be utilized by these compounds to suppress VZV and HCMV replication, which will be the focus of our future investigations. In addition to inhibiting cellular-signaling pathways and transcription factors for MIEP activation, quercetin and isoquercitrin may also interfere with virus entry and uncoating or nucleocapsid entry into the nucleus.

## 4. Materials and Methods

### 4.1. Cells, Viruses, and Chemicals

The maintenance and propagation of primary HFF cells, the recombinant laboratory pOka strain of VZV (VZV–pOka), and the Towne strain of HCMV (HCMV–Towne) were previously described [39,40]. Digallic acid was purchased from Santa Cruz Biotechnology (Dallas, TX, USA). Luteolin-7-rutinoside, isoquercitrin, quercetin-3-O-arabinoside, quercetin, galloyl-D-glucose, gallic acid, trigalloyl glucose, ellagic acid, 1,2,3,4,6-penta-O-galloyl-ß-D-glucose, and acyclovir (ACV) were purchased from Sigma-Aldrich (St. Louis, Mo, USA). Ganciclovir (GCV) was purchased from Tokyo Chemical Industry (Tokyo, Japan).

### 4.2. Plaque-Reduction Assay

HFF cells were inoculated with serially diluted VZV–pOka-infected HFF cells or infected with serially diluted HCMV–Towne and treated with DMSO, quercetin, or isoquercitrin at concentrations of 0, 1, 5, 10, 20, and 50 µg/mL. Cells were retreated with DMSO, quercetin, or isoquercitrin every 3 days. For VZV experiments, cells were fixed and stained at 6 days after infection as described previously [5]. For HCMV experiments, cells were agar-overlaid at 3 and 10 days after infection. At 14 days postinfection, cells were fixed with 10% formaldehyde for 10 min at room temperature and stained with 0.03% methylene blue. The number of plaques was counted, and virus titers expressed as plaque-forming units (pfu/mL).

### 4.3. Cell-Viability Assay

Cell viability was determined using a CellTiter-Glo^®^ luminescent cell-viability assay according to the manufacturer’s instructions (Promega Corporation Madison, WI, USA).

### 4.4. Quantification of Viral DNA and RNA

For quantitative analysis of viral DNA and RNA transcripts, quantitative polymerase chain reaction (qPCR) and quantitative reverse transcription PCR (RT-PCR) were performed as described previously [5,41]. The primer sequences used for amplification were as follows: VZV *ORF62* (IE) forward 5′-TCTTGTCGAGGAGGCTTCTG-3′ and reverse 5′ -TGTGTGTCCACCGGATGAT- 3′; VZV *ORF28* (E) forward 5′ -CGAACACGTTCCCCATCAA-3′ and reverse 5′ -CCCGGCTTTGTTAGTTTTGG- 3′; VZV *gB* (L) forward 5′-GATGGTGCATACAGAGAACATTCC-3′ and reverse 5′ -CCGTTAAATGAGGCGTGACTAA- 3′; HCMV *UL123* forward 5′-GCCTTCCCTAAGACCACCAAT-3′ and reverse 5′ -ATTTTCTGGGCATAAGCCATAATC- 3′; HCMV *UL122* (IE) forward 5′-ACCATGCAGGTGAACAACAA-3′ and reverse 5′ -CATGAGGAAGGGAGTGGAGA- 3′; HCMV *UL44* (E) forward 5′-GCTGTCGCTCTCCTCTTTCG-3′ and reverse 5′ -TCACGGTCTTTCCTCCAAGG- 3′; HCMV *UL83* (L) forward 5′-GCAGCCACGGGATCGTACT-3′ and reverse 5′- GGCTTTTACCTCACACGAGCATT-3′; *GAPDH* forward 5′-CATGAGAAGTATGACAACAGCCT-3′ and reverse 5′ -AGTCCTTCCACGATACCAAAGT- 3′.

### 4.5. Western Blot Analysis

Cells were harvested, fractionated, and transferred onto nitrocellulose membranes as described previously [42]. Antibodies specific for VZV IE62 were purchased from Abcam (Cambridge, UK), HCMV IE (IE2-86kDa) from Virusys (Taneytown, MD, USA), and tubulin from Sigma-Aldrich (St. Louis, MO, USA).

## Figures and Tables

**Figure 1 molecules-25-02379-f001:**
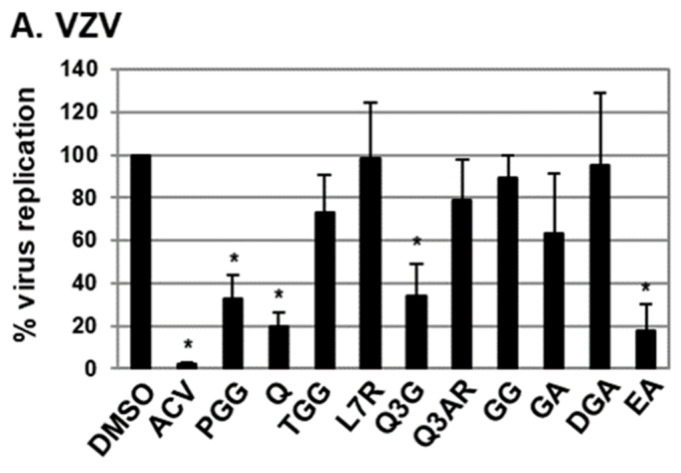
Antiviral activities of chemical components of ethyl acetate (EtOAc) fraction of *Elaeocarpus sylvestris* (ESE). HFF cells were (**A**) inoculated with varicella-zoster-virus (VZV)–recombinant laboratory pOka strain (pOka)-infected HFF cells or (**B**) infected with human cytomegalovirus (HCMV)–Towne strain (Towne) at an multiplicity of infection (MOI) of 0.1 and treated with DMSO, 1,2,3,4,6-penta-O-galloyl-ß-D-glucose (PGG), quercetin (Q), trigalloyl glucose (1,3,6-tri-*O*-galloyl- β-D-glucose, TGG), luteolin-7-rutinoside (L7R), isoquercitrin (quercetin 3-*O*-β-D-glucoside, Q3G), quercetin-3-O-arabinoside (Q3AR), galloyl-D-glucose (1-*O*-galloyl-β-D-glucose, GG), gallic acid (GA), digallic acid (DGA), and ellagic acid (EA) at a concentration of 20 µg/mL. As a control, VZV- and HCMV-infected HFF cells were treated with acyclovir (ACV, 1.2 µg/mL) and ganciclovir (GCV, 2.3 µg/mL), respectively. At 72 h after infection, total DNA was harvested, and relative amounts of viral DNA determined via qPCR using primers specific for VZV ORF62 or HCMV UL123. For quantitative assessment of the relative abundance of viral DNA between samples, the value of virus-infected cells treated with DMSO was set at 100. Significant differences between samples were determined using Student’s *t* test (significant at **p* < 0.05). Data shown here represent three independent sets of experiments.

**Figure 2 molecules-25-02379-f002:**
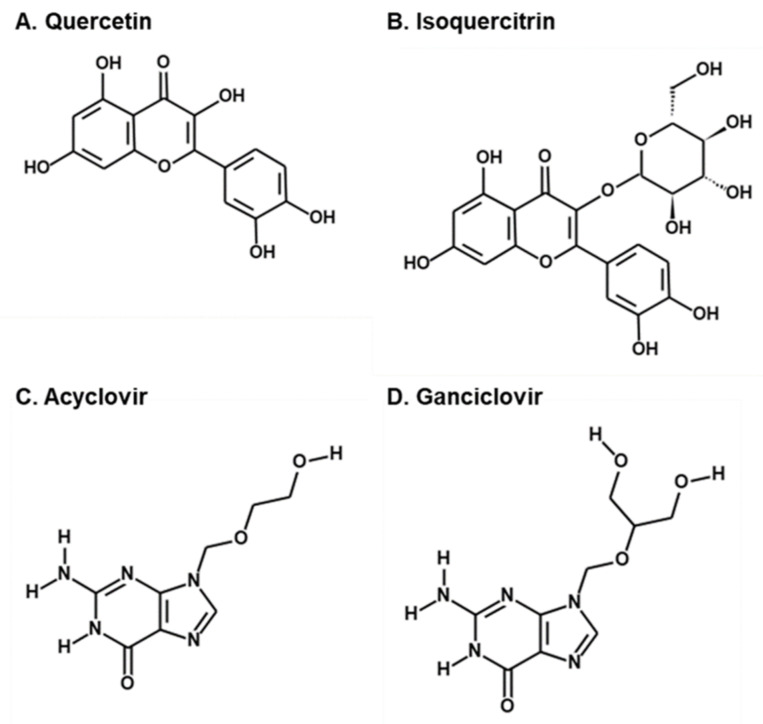
Structures of (**A**) quercetin, (**B**) isoquercitrin, (**C**) acyclovir, and (**D**) ganciclovir [19,20].

**Figure 3 molecules-25-02379-f003:**
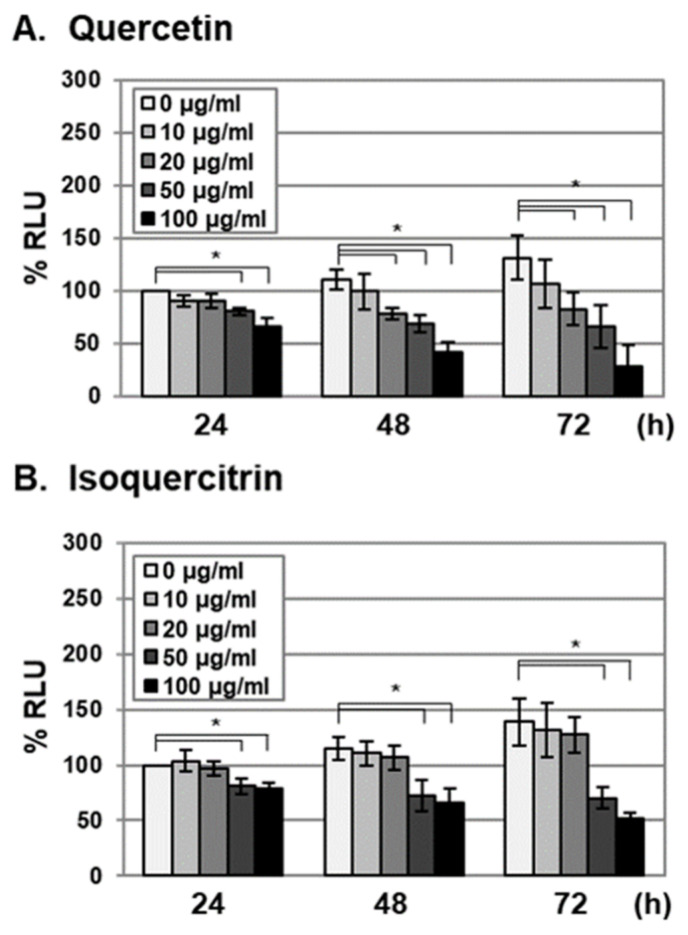
Cytotoxic effects of quercetin and isoquercitrin against HFF cells. HFFs were treated with (**A**) quercetin or (**B**) isoquercitrin at concentrations of 0, 10, 20, 50, and 100 µg/mL. At 24, 48, and 72 h after treatment, cell viability was determined by measuring cellular ATP levels using the CellTiter-Glo^®^ Luminescent cell-viability assay. Significant differences between samples were determined using Student’s *t* test (significant at **p* < 0.05). Data shown here represent three independent sets of experiments. RLU, relative luciferase unit.

**Figure 4 molecules-25-02379-f004:**
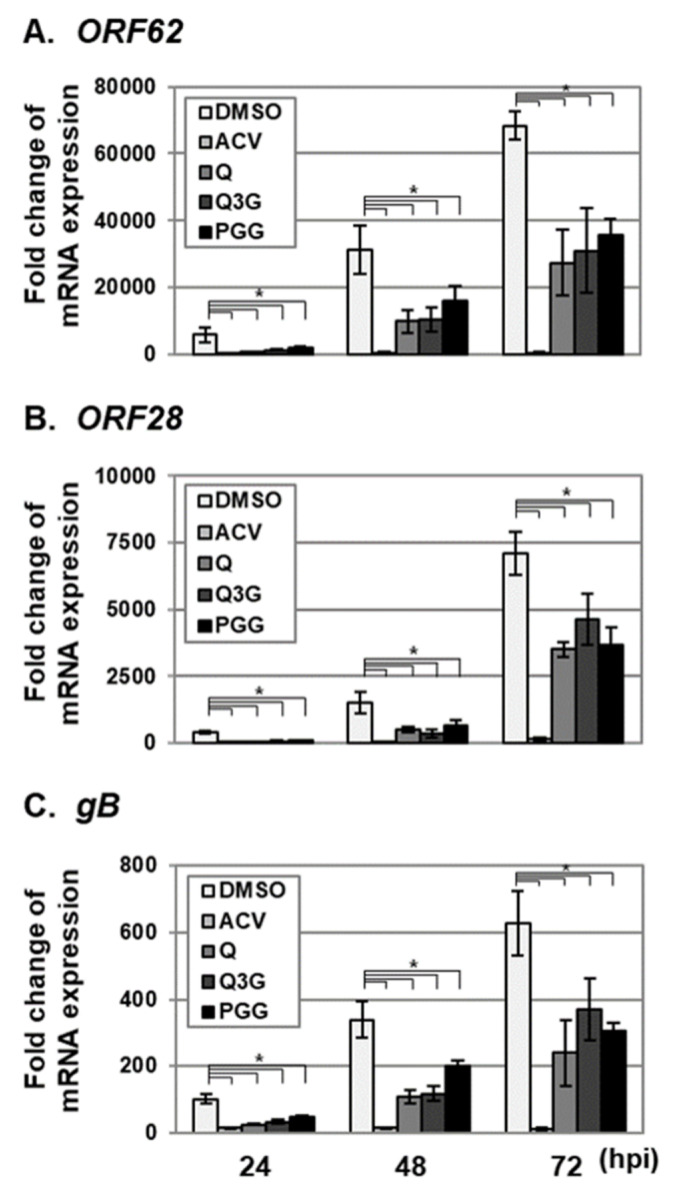
Effects of quercetin and isoquercitrin on VZV lytic-gene expression. HFF cells were inoculated with VZV–pOka-infected HFF cells at an MOI of 0.1, followed by treatment with DMSO, quercetin (Q), and isoquercitrin (Q3G) or 1,2,3,4,6-penta-O-galloyl-ß-D-glucose (PGG) at a concentration of 20 µg/mL. As a control, VZV–pOka-infected HFF cells were treated with ACV (1.2 µg/mL). At 24, 48, and 72 h after infection, total RNA was isolated and reverse-transcribed into cDNA. Relative expression of (**A**) *ORF62* (IE), (**B**) *ORF28* (E) and (**C**) *gB* (L) mRNA was assessed via qRT–PCR. For quantification of the relative abundance of viral mRNA between samples, the value of virus-infected cells treated with DMSO was set at 100. Significant differences between samples were determined using Student’s *t* test (significant at **p* < 0.05). Data shown here represent three independent sets of experiments.

**Figure 5 molecules-25-02379-f005:**
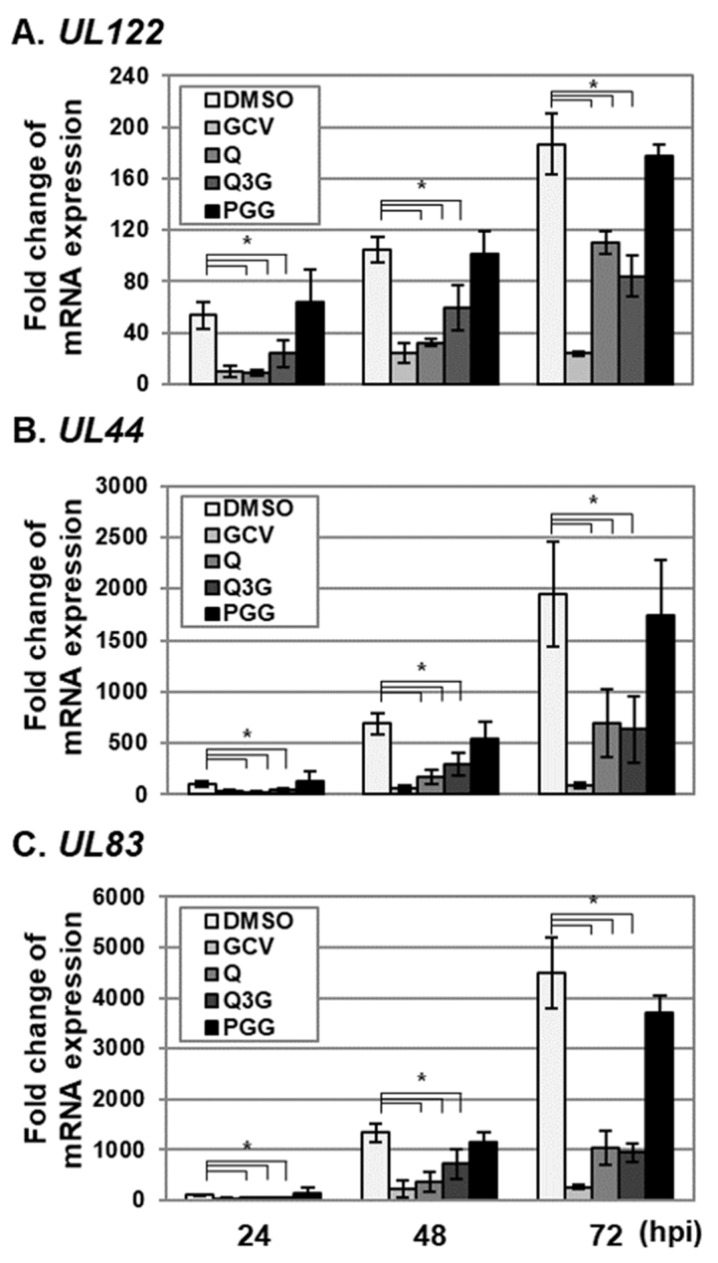
Effects of quercetin and isoquercitrin on HCMV lytic-gene expression. HFF cells were infected with HCMV–Towne at an MOI of 0.1, followed by treatment with DMSO, quercetin (Q), and isoquercitrin (Q3G) or 1,2,3,4,6-penta-O-galloyl-ß-D-glucose (PGG) at a concentration of 20 µg/mL. As a control, HCMV–Towne-infected HFF cells were treated with GCV (2.3 µg/mL). At 24, 48, and 72 h after infection, total RNA was isolated and reverse-transcribed into cDNA. Relative mRNA levels of (**A**) *UL122* (IE), (**B**) *UL44* (E), and (**C**) *UL83* (L) were measured via qRT–PCR. For quantification of the relative abundance of viral mRNA between samples, the value of virus-infected cells treated with DMSO was set at 100. Significant differences between samples were determined using Student’s *t* test (significant at **p* < 0.05). Data shown here represent three independent sets of experiments.

**Figure 6 molecules-25-02379-f006:**
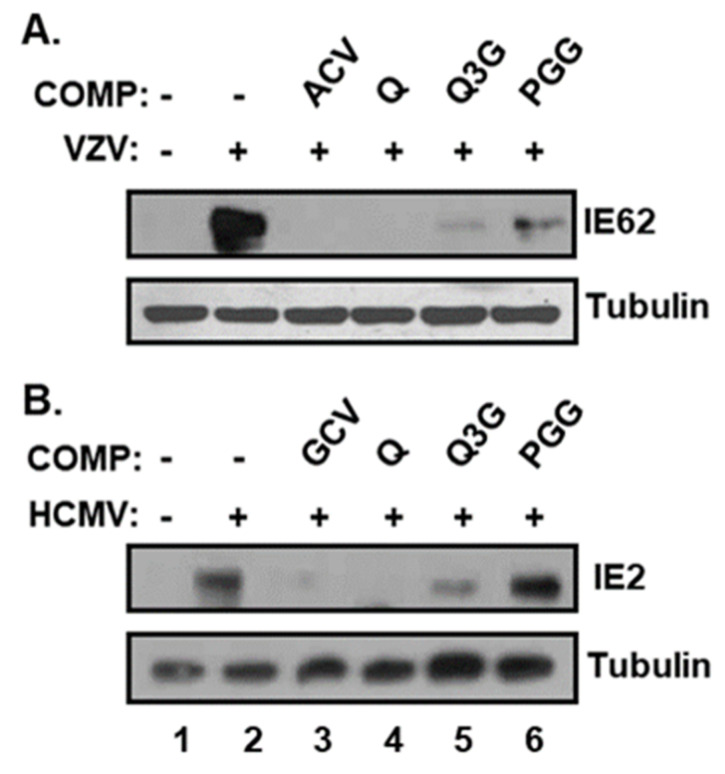
Effects of quercetin and isoquercitrin on VZV and HCMV IE protein levels. HFF cells were (**A**) inoculated with VZV–pOka-infected HFF cells or (**B**) infected with HCMV–Towne at an MOI of 0.1, followed by treatment with DMSO, quercetin (Q), and isoquercitrin (Q3G) or 1,2,3,4,6-penta-O-galloyl-ß-D-glucose (PGG) at a concentration of 20 µg/mL. As a control, VZV- and HCMV-infected HFF cells were treated with ACV (1.2 µg/mL) and GCV (2.3 µg/mL), respectively. At 48 h after infection, equal amounts of cell lysates were subjected to Western blot with antibodies against VZV IE62, HCMV IE2, and tubulin. COMP, compound.

**Table 1 molecules-25-02379-t001:** Average 50% inhibitory-concentration (IC_50_) values for antiviral activities of quercetin and isoquercitrin against VZV and HCMV.

Virus	IC_50_(μg/mL)	
	Quercetin	Isoquercitrin
VZV	3.835 ± 0.56	14.4 ± 2.77
HCMV	5.931 ± 1.195	1.852 ± 1.115

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
