# Peer review of "Antiviral Activities of Quercetin and Isoquercitrin Against Human Herpesviruses"

_molecules, 2020, doi:10.3390/molecules25102379_

Round 1
Reviewer 1 Report
In this study, Chae Hyun Kim and colleagues show that quercetin and isoquercitrin have potent anti-viral activity against VZV and HCMV in an in vitro setting.The experiments are well conducted and easy to follow and the conclusions are valid. Below are a few suggestions:
The study would be strengthen by studying the effects of the selected compounds when administered a few days post viral infection. This is crucial from the perspective of designing a therapeutic treatment.
The discussion need to be expanded and include the vast litterature on the mechanistic effects of quercetin and isoquercitrin on other herpesviruses, e.g. HSV.
Author Response
To: Mr. Males,
Assistant Editor, Molecules
Dear Mr. Males,
Please find below our response to the reviewers for the manuscript # molecules-790382 (Title: Antiviral activities of quercetin and isoquercitrin against human herpesviruses) item by item. We appreciate the comments of the reviewers and your assistance.
Best regards,
Yoon-Jae Song, Ph.D.
Response to the reviewer’s comments:
Reviewer 1:
In this study, Chae Hyun Kim and colleagues show that quercetin and isoquercitrin have potent anti-viral activity against VZV and HCMV in an in vitro setting.The experiments are well conducted and easy to follow and the conclusions are valid. Below are a few suggestions:
1) The study would be strengthen by studying the effects of the selected compounds when administered a few days post viral infection. This is crucial from the perspective of designing a therapeutic treatment.
- As suggest by the reviewer, we performed post-treatment study and found out that 24 and 48 h post-treatment of quercetin and isoquercitrin had no significant antiviral effects against VZV and HCMV. Chiang et al (J Antimicrob Chemother 2003, 52, (2), 194-8) also reported that quercetin inhibits HSV-1 infection in the early stages of the viral life cycle (0 to 2 h post-infection). This information is included in the discussion section (lines 194-200).
2) The discussion need to be expanded and include the vast litterature on the mechanistic effects of quercetin and isoquercitrin on other herpesviruses, e.g. HSV.
- As suggested by the reviewer, the discussion section is expanded to include the studies about antiviral activities of quercetin and isoquercitrin against HSV (lines 190-195).
Reviewer 2 Report
Chae Hyun Kim et al previously reported that herpesviruses varicella-zoster virus (VZV) and human cytomegalovirus (HCMV) replication is inhibited by the ethyl acetate fraction of a 70% ethanol extract of Elaeocarpus sylvestris, and identified 13 compound present in this extract.
In this paper the authors investigate which compound are responsible for the inhibitory effect and attempt to understand by which mechanism. They observe quercetin and isoquercitrin have a robust antiviral effect on both viruses which results in down expression of viral genes.
The paper is well written, and the figures are clear. The overall strategy makes sense, the authors initiate the investigation of the mechanism by which quercetin and isoquercitrin inhibits the two herpesviruses.
It is unclear why the authors do not pursue the study of ellagic acid (one of the 13 initial compounds tested), in fact this compound has a similar inhibition effect to quercetin and isoquercitrin based on the data provided in figure 1, therefore a justification of this decision appears necessary.
Also, in the discussion the authors suggest that quercetin and isoquercitrin inhibit viral lytic gene expression and replication through down-regulation of VZV and HCMV immediate early genes, and thus may inhibit – directly or indirectly – the transcription of viral genes. The same down expression would likely be observed if the virus entry and viral DNA nuclear import were inhibited by quercetin and isoquercitrin, it would be beneficial for the paper to also discuss that possibility.
Altogether the paper identifies two interesting compounds to inhibit herpesviruses VZV and HCMV in cellulo and match the scope of Molecules journal.
Specific comments
line 71. Does this system mimics primary infections or reactivation? A sentence to describe the infection system used would be useful to the readers.
Line 79. Please explain why EA was not kept in this study.
Line 88-89. I believe there is a typo in ACV and GCV concentration unit?
Line 93. Please specify what the error bar stands for and numbers of repeats.
Line 95. Insert reference. Suggestion: include ACV and GCV for comparison.
Line 103. Table1. What is the IC50 of known effective antiviral ACV and GCV in this specific assay?
Figure3. Please add 24, 48 and 72h under graph in panel A.
Line 127. Reference missing
Line 131. Reference missing
Line 138. Unit typo?
Line 142. Specify what the error bar stands for and numbers of repeats.
Line 147. Unit typo?
Line 152. Specify what the error bar stands for and numbers of repeats.
Line 153. For better clarity, perhaps state that the level of mRNA of previously tested genes are low, suggesting the effect of the compound occurs before that step and therefore you looked into these proteins.
Line 154. Extend the introduction of these two proteins and insert reference.
Line 163. Inoculated with VZV-pOka instead of VZV-pOka-infected HFF cells.
Line 166. Unit typo?
Line 185-190. Please include the possibility to inhibit other steps such as viral entry and nuclear import. Unless there are evidence supporting against it, then include that in the discussion.
Line 202. Missing closing parenthesis.
Line 220. Was this done in replicates?
Line 235. Was this done in replicates?
Author Response
To: Mr. Males,
Assistant Editor, Molecules
Dear Mr. Males,
Please find below our response to the reviewers for the manuscript # molecules-790382 (Title: Antiviral activities of quercetin and isoquercitrin against human herpesviruses) item by item. We appreciate the comments of the reviewers and your assistance.
Best regards,
Yoon-Jae Song, Ph.D.
Response to the reviewer’s comments:
Reviewer 2:
Chae Hyun Kim et al previously reported that herpesviruses varicella-zoster virus (VZV) and human cytomegalovirus (HCMV) replication is inhibited by the ethyl acetate fraction of a 70% ethanol extract of Elaeocarpus sylvestris, and identified 13 compound present in this extract.
In this paper the authors investigate which compound are responsible for the inhibitory effect and attempt to understand by which mechanism. They observe quercetin and isoquercitrin have a robust antiviral effect on both viruses which results in down expression of viral genes.
The paper is well written, and the figures are clear. The overall strategy makes sense, the authors initiate the investigation of the mechanism by which quercetin and isoquercitrin inhibits the two herpesviruses.
1) It is unclear why the authors do not pursue the study of ellagic acid (one of the 13 initial compounds tested), in fact this compound has a similar inhibition effect to quercetin and isoquercitrin based on the data provided in figure 1, therefore a justification of this decision appears necessary.
- As described in results, ellagic acid exerted a significant cytotoxic effect on HFF cells (lines 76-78). Thus, we decided to focus on quercetin and isoquercitrin.
2) Also, in the discussion the authors suggest that quercetin and isoquercitrin inhibit viral lytic gene expression and replication through down-regulation of VZV and HCMV immediate early genes, and thus may inhibit – directly or indirectly – the transcription of viral genes. The same down expression would likely be observed if the virus entry and viral DNA nuclear import were inhibited by quercetin and isoquercitrin, it would be beneficial for the paper to also discuss that possibility.
- As suggested by the reviewer, we revised the discussion section to address the possibility that quercetin and isoquercitrin interfere with other steps in viral life cycle (lines 214-216).
Altogether the paper identifies two interesting compounds to inhibit herpesviruses VZV and HCMV in cellulo and match the scope of Molecules journal.
Specific comments
1) line 71. Does this system mimics primary infections or reactivation? A sentence to describe the infection system used would be useful to the readers.
- As described in the legend for figure 1, HFF cells were infected with 0.1 MOI of viruses (lines 83-88).
2) Line 79. Please explain why EA was not kept in this study.
- As described in results, ellagic acid exerted a significant cytotoxic effect on HFF cells (lines 76-78).
3) Line 88-89. I believe there is a typo in ACV and GCV concentration unit?
- Typos are corrected.
4) Line 93. Please specify what the error bar stands for and numbers of repeats.
- The information requested by the reviewer is included in a legend for figure 1 (lines 93-95) as follows: “Significant differences between samples were determined using Student’s t test (significant at * P < 0.05). The data shown here represent three independent sets of experiments.”
5) Line 95. Insert reference. Suggestion: include ACV and GCV for comparison.
- Changes were made as suggested by the reviewer.
6) Line 103. Table1. What is the IC50 of known effective antiviral ACV and GCV in this specific assay?
- As requested by the reviewer, the information is included in lines 100-101 as follows: “The average 50% inhibitory concentrations (IC50) of ACV for VZV and GCV for HCMV are 3 µg/ml and 0.89 µg/ml, respectively [20, 21].”
7) Figure3. Please add 24, 48 and 72h under graph in panel A.
- Changes were made as suggested by the reviewer.
8) Line 127. Reference missing
- A reference is included. As described in lines 131-132, ORF62 is VZV immediate-early (IE) gene, ORF28 is VZV early (E) gene and gB is VZV late (L) gene.
9) Line 131. Reference missing
- A reference is included. As described in lines 134-135, UL122 is HCMV immediate-early (IE) gene, UL44 is HCMV early (E) gene and UL83 is HCMV late (L) gene.
10) Line 138. Unit typo?
- A typo is corrected.
11) Line 142. Specify what the error bar stands for and numbers of repeats.
- The information requested by the reviewer is included in a legend for figure 4 (lines 145-147) as follows: “Significant differences between samples were determined using Student’s t test (significant at * P < 0.05). The data shown here represent three independent sets of experiments.”
12) Line 147. Unit typo?
- A typo is corrected.
13) Line 152. Specify what the error bar stands for and numbers of repeats.
- The information requested by the reviewer is included in a legend for figure 4 (lines 156-157) as follows: “Significant differences between samples were determined using Student’s t test (significant at * P < 0.05). The data shown here represent three independent sets of experiments.”
14) Line 153. For better clarity, perhaps state that the level of mRNA of previously tested genes are low, suggesting the effect of the compound occurs before that step and therefore you looked into these proteins.
- As described in lines 158-160, we confirmed the qRT-PCR data (reduction of ORF62 and UL122 transcripts) with western blot analysis (reduction of IE62 encoded by ORF62 and IE2 encoded by UL122). VZV ORF62 gene encodes IE62 protein, and HCMV UL122 gene encodes IE2 protein.
15) Line 154. Extend the introduction of these two proteins and insert reference.
- VZV IE62 and HCMV IE2 are described in lines 158-160, and a reference is included as suggested by the reviewer.
16) Line 163. Inoculated with VZV-pOka instead of VZV-pOka-infected HFF cells.
- As described in introduction (lines 48-49), VZV is highly cell-associated and spreads via cell-to-cell contact. Thus, VZV infection was performed by inoculating un-infected cells with VZV-pOka-infected cells.
17) Line 166. Unit typo?
- Typos are corrected.
18) Line 185-190. Please include the possibility to inhibit other steps such as viral entry and nuclear import. Unless there are evidence supporting against it, then include that in the discussion.
- As suggested by the reviewer, we revised the discussion section to address the possibility that quercetin and isoquercitrin interfere with other steps in viral life cycle (lines 214-216).
19) Line 202. Missing closing parenthesis.
- A typo is corrected.
20) Line 220. Was this done in replicates?
- As described in figure legends, the quantitative data in this paper represent the average of three independent sets of experiments.
21) Line 235. Was this done in replicates?
- As described in figure legends, the quantitative data in this paper represent the average of three independent sets of experiments.
Round 2
Reviewer 1 Report
OK.